# From Mice to Humans: An Overview of the Potentials and Limitations of Current Transgenic Mouse Models of Major Muscular Dystrophies and Congenital Myopathies

**DOI:** 10.3390/ijms21238935

**Published:** 2020-11-25

**Authors:** Mónika Sztretye, László Szabó, Nóra Dobrosi, János Fodor, Péter Szentesi, János Almássy, Zsuzsanna É. Magyar, Beatrix Dienes, László Csernoch

**Affiliations:** Department of Physiology, Faculty of Medicine, University of Debrecen, H-4002 Debrecen, Hungary; sztretye.monika@med.unideb.hu (M.S.); laszlo.szabo@med.unideb.hu (L.S.); dobrosi.nora@med.unideb.hu (N.D.); fodor.janos@med.unideb.hu (J.F.); szentesi.peter@med.unideb.hu (P.S.); almassy.janos@med.unideb.hu (J.A.); magyar.zsuzsa@med.unideb.hu (Z.É.M.); dienes.beatrix@med.unideb.hu (B.D.)

**Keywords:** mouse models, muscle disorders, dystrophy, dystrophinopathies, myopathy, malignant hyperthermia

## Abstract

Muscular dystrophies are a group of more than 160 different human neuromuscular disorders characterized by a progressive deterioration of muscle mass and strength. The causes, symptoms, age of onset, severity, and progression vary depending on the exact time point of diagnosis and the entity. Congenital myopathies are rare muscle diseases mostly present at birth that result from genetic defects. There are no known cures for congenital myopathies; however, recent advances in gene therapy are promising tools in providing treatment. This review gives an overview of the mouse models used to investigate the most common muscular dystrophies and congenital myopathies with emphasis on their potentials and limitations in respect to human applications.

## 1. Introduction

Muscle dystrophy is a muscle disease that leads to a progressive loss of muscle mass and a weakened musculoskeletal system in accordance with age of onset, severity, and the group of muscles affected. Dystrophy is an umbrella name that encompasses more than 30 genetic disorders that progress over time, leading to degeneration and weakness of the muscles. The phenotype of muscular dystrophy is an endpoint that arises from a disparate set of genetic and biochemically heterogeneous pathways. Genes associated with muscular dystrophies encode proteins of the plasma membrane (sarcolemma), terminal cisternae, extracellular matrix, and the sarcomere, as well as nuclear membrane components (Figure 1).

Myopathies are a diversified family of disorders characterized by pathological structure and/or the functioning of skeletal muscles. Inherited myopathies include a clinically, histopathologically, and genetically heterogeneous group of rare genetic muscle diseases that are characterized by architectural anomalies in the muscle fibers.

In the present review, our aim was to focus primarily on the mouse models used in preclinical studies of the amplest muscle disorders with emphasis on their potentials and limitations in respect to human applications. Here, we chose to elaborate exclusively on mouse models as they are easy to breed, maintain in large numbers, and genetically modify; however, one must note that there are various mammalian model systems available that are not addressed here due to length constraints. The importance of establishing similarities and differences between the human disease condition and murine animal models and the potential obstacles and limitations that arise from these differences when attempting to elucidate a prospective therapeutic strategy for muscle disorders is now generally accepted. Even though scientists have accesses to robust methods for the diagnosis and extensive characterization of disease progression along with a vast array of animal models that recapitulate well (but not entirely) muscle disorders, the available therapies are still palliative, minimizing symptoms rather than addressing the true cause of the disease. 

## 2. Muscular Dystrophies 

Muscular dystrophies (MDs) are a group of inherited disorders in which the voluntary muscles that control movement, in some instances the heart muscles and eventually the diaphragm, progressively weaken and lose their ability to maintain proper function. There are more than 30 types of MDs that vary in severity, symptoms, and causes. In recent years, the classification of MDs has been adjusted in order to correspond to the newly available information related to the primary protein dysfunctions and their localizations. As a consequence, by convention, the MDs had been classified according to the main clinical and biopsy findings, age of onset, and rate of progression into nine major forms: (1) Becker, (2) congenital, (3) Duchenne, (4) distal, (5) Emery–Dreifuss, (6) facioscapulohumeral, (7) limb–girdle, (8) myotonic, and (9) oculopharyngeal muscular dystrophy. In the present review, we tackle the most common forms of MDs in humans. 

### 2.1. Dystrophinopathies

Dystrophinopathies cover a spectrum of X-linked muscle diseases ranging from mild to severe forms that include Duchenne muscular dystrophy (DMD), Becker muscular dystrophy (BMD), and DMD-associated dilated cardiomyopathy (DCM). DMD/BMD are neuromuscular genetic disorders characterized by progressive muscle degeneration, weakness, and wasting due to the alterations of a critical muscle protein called dystrophin, which is a relatively long (110 nm), rod-shaped intracellular protein localized at the cytoplasmic face of the sarcolemma in cardiac and skeletal muscles [1,2]. Dystrophin connects γ-actin of the subsarcolemmal cytoskeleton system to a complex of proteins in the surface membrane (dystrophin protein complex, DPC) and helps keep the muscle cell intact and orchestrates the transmission of force laterally across the muscle during contraction (see Figure 1). Growing evidence suggests that dystrophin also has a major role in regulating signaling pathways that activate nitric oxide (NO) production, Ca^2+^ entry, and the production of reactive oxygen species (ROS). The absence or reduced expression of dystrophin beside other members of the DPC complex causes dystrophinophathies.

DMD is the most common muscular dystrophy in children affecting primarily boys due to classical X-linked recessive genetics according to which males who carry the mutation express the disease while females are carriers. The incidence of DMD is approximately 1 in 3500 [3,4,5,6]. DMD symptom onset is in early childhood, usually between ages 2 and 6. The symptoms of DMD include decreased muscle size accompanied by progressive weakness and atrophy of skeletal and heart muscles. Early signs of DMD are muscle weakness of weight-bearing muscles and may include delayed ability to sit, stand, or walk, difficulties learning to speak, and general cognitive impairment. Most children with DMD use a wheelchair by their early teens. Heart and breathing problems also begin in the teen years, leading to serious life-threatening complications, and patients usually die in the third or fourth decade due to respiration or cardiac failure.

Frame-shift mutations or other genetic rearrangement in the dystrophin gene abolish protein expression that disturbs the connection between the cytoskeleton and the extracellular matrix, making muscle fibers more susceptible to contraction-induced membrane damage. As a result, the uncontrolled influx of calcium ions occurs inevitably, leading to progressive myofiber degeneration [7,8]. These pathologic processes are accompanied by chronic inflammation and fibrosis [9], as evidenced by macrophage infiltration [10]. In DMD, skeletal muscles active myofiber necrosis, and cellular infiltration can be histologically identified, furthermore regenerating myofibers containing centrally located nuclei, and a large variety of myofiber sizes are often detected. This phenotype is particularly pronounced in the diaphragm, which undergoes progressive degeneration and myofiber loss, causing an approximately 5-fold reduction in muscle isometric strength [11].

In-frame deletions often generate truncated dystrophin and result in BMD characterized by skeletal muscle weakness with milder symptoms and later onset that appear between the ages of 2 and 16 but in some cases as late as the twenties. Its incidence has been estimated to be between 1 in 30,000 male births [12,13].

Plentiful mouse models have been developed to better understand the basic molecular biology of DMD. Currently, there are nearly 60 different animal models for DMD, and the list keeps growing. For a comprehensive lineup, see Table 1 and the review by McGreevey and colleagues [14]. The TREAT-NMD Alliance (https://treat-nmd.org/research-overview/preclinical-research/) is an initiative to improve preclinical trial design and execution for the most common mouse models of DMD, spinal muscular atrophy (SMA), and congenital muscular dystrophy.

With all its caveats, the most widely used animal model for DMD research is the C57BL/10ScSn-Dmdmdx/J (BL10-mdx; available from the Jackson laboratory, JL#001801) mouse in which the dystrophic phenotype arises because of a point mutation (C to T transition) in exon 23, which results in a stop codon and truncated dystrophin protein. This spontaneous mutation was discovered in the early 1980s in a colony of C57BL/10ScSn mice due to elevated serum creatine kinase (CK) and histological evidence of myopathy [15]. The mdx muscles seem more susceptible to contraction- and stretch-induced damage revealed as sarcolemmal tears [64]. Normal physiological control of calcium homeostasis is lost in mdx mice [65,66], and similar to the human condition, calcium levels are increased in myofibers isolated from mdx mice [67].

DMD is a multi-systemic condition affecting many parts of the body and resulting in atrophy of the skeletal, cardiac, and respiratory muscles. DMD disease progression in mdx mice has several distinctive phases. In the first 2 weeks, the mdx muscle is indistinguishable from that of normal mice. Between 3 and 6 weeks, it undergoes astonishing necrosis. Subsequently, the majority of skeletal muscle enters a relatively robust regeneration phase. As a hallmark of the disease, mdx limb muscles often become hypertrophic during this phase. The diaphragm is an exception, as it shows progressive deterioration, as seen in affected humans [11]. Severe dystrophic phenotypes, such as muscle wasting, scoliosis, and heart failure do not occur until mdx mice are 15 months or older [68,69,70,71,72,73]. Despite being deficient for dystrophin, mdx mice display overall minimal clinical symptoms; their lifespan is only reduced by ≈25% (vs. 75% decrease in humans) without obvious signs of dilated cardiomyopathy [14,37]. The robust skeletal muscle regeneration might explain somewhat the slowly progressive phenotype observed in mdx mice.

The mdx mouse has been crossed to several different genetic backgrounds, including the Albino, BALB/c, C3H, C57BL/6, DBA2, and FVB strains; several immune-deficient mdx strains were also engineered (see Table 1). Phenotypic variation has been observed in different backgrounds. Several other dystrophin-deficient lines (Dup2, DMD-null, Dp71-null, mdx52, and mdx^βgeo^) were also created using various genetic engineering techniques. The DMD-null mouse was created by deleting the entire DMD genomic region using the Cre-loxP technology [24] resulting in the ablation dystrophin isoforms expression in all tissues. Further models (mdx^cv^) were created by chemical mutagenesis programs by treating mice with N-ethyl-N-nitrosourea, a chemical mutagen, so that each strain carries a different point mutation [20,74]. By eliminating myogenic differentiation 1 (MyoD), a master myogenic regulator, from mdx mice, Megeney et al. obtained a MyoD/dystrophin double-mutant mouse that shows marked myopathy, dilated cardiomyopathy, and premature death [45,46,75]. Another similar approach was the generation of telomerase/mdx double-mutant mice (mTR/mdx) that show more severe muscle wasting and cardiac defects [44,76].

Two other proteins, utrophin and α7-integrin, fulfill the same function as dystrophin, and their relative expression is upregulated in mdx mice. The genetic elimination of utrophin, which is expressed along the sarcolemma in developing muscle, exhibits 80% homology and shares structural and functional motifs with dystrophin and α7-integrin; their deletion in mdx mice lead to the creation of utrophin/dystrophin and integrin/dystrophin double-knockout (dko) mice, respectively [33,34,35,46,77]. The dko mice show much more severe muscle disease symptoms (similar to or even worse than that of humans with DMD); however, they are difficult to generate and care for. Utrophin heterozygous mdx mice might represent an intermediate model between the extreme dko mice and mildly affected mdx mice [78,79].

Second mutations have been introduced to “humanize” mice (e.g., inactivation of cytidine monophosphate sialic acid hydrolase (Cmah)) and to mutate genes involved in cytoskeleton-ECM interactions (e.g. desmin and laminin); however, the introduction of a second mutation not present in human DMD turned out to produce a much more severe phenotype and complicated data interpretation [14,36,46,80].

To test if the “humanization” of telomere lengths could recapitulate the DMD disease phenotype, the mdx^4cv^/mTR^G2^ dko mice were generated, which seem to recapitulate the best of both the skeletal muscle and cardiovascular features of human DMD [44]. Nevertheless, there are still a few tenable therapies for DMD, so the need for appropriate mouse models more similar to the mdx model is emphasized. Even with an improved delivery of promising strategies such as gene editing or exon skipping, testing must be done in mice with the full spectrum of DMD pathology.

Dysferlinopathies are caused by the lack of functional dysferlin, which is a key protein involved in membrane repair processes causing Myoshi myopathy or dysferlin-related limb girdle muscular dystrophy (LGMD R2) [81]. The dysferlin-deficient mice (dysf^-/-^) replicate well human dysferlinopathies, showing similarities with the human condition although with milder histopathological aspects. Due to space restrictions, we did not further elaborate on these mouse models (for a comprehensive review, see [82] and more recently [83].

### 2.2. Myotonic Dystrophy

Myotonic dystrophy (DM) is an autosomal dominantly inherited disorder and the most prevalent form of muscular dystrophy in adulthood. Clinical characterization of the disease was done first by Steiner in 1909. DM is a complex genetic disease with diverse symptoms affecting multiple organs, such as skeletal muscle, cardiac muscle, the endocrine and gastrointestinal system, reproductive system, and central nervous system (CNS). Symptoms range from muscle weakness and wasting both in skeletal muscle and in heart, arrhythmias, or conduction abnormalities, disorders in the function of the neuromuscular junction, neurologic impairment such as excessive daytime sleepiness and motivation deficit, insulin resistance, cataracts, and male infertility. There are two major forms of the disease: myotonic dystrophy type I (DM1 or Steiner’s disease) and myotonic dystrophy type II (DM2 or proximal myotonic myopathy), which are associated to partially similar clinical appearances but distinct genetic defects [57].

Several hypotheses have been suggested to explain the complex symptoms of DM. The genetic background responsible for classic myotonic dystrophy documented by Steiner was discovered in 1992. An expansion of a CTG trinucleotide repeat in the 3′ untranslated region of the dystrophia myotonica protein kinase gene (DMPK) has been identified, which is a mutation that has been transcribed into RNA but not translated into protein [57]. Based on DMPK haploinsufficiency theory, the expanded repeats inhibit DMPK mRNA or protein production, which is in agreement with observation in DM1 patient muscle and cell cultures demonstrating a decreased expression of DMPK mRNA and protein [84]. On the other hand, DMPK-knockout mice did not display myotonia but rather mild myopathy [85]. Although DMPK haploinsufficiency alone is not sufficient to explain the features of DM1, the CTG repeats might influence the expression of neighboring genes as well. The haploinsufficiency of SIX5 and of other adjacent genes such as myotonic dystrophy gene with tryptophan and aspartic acid (WD) repeats, DMWD [86], and the FCGRT gene, encoding the Immunoglobulin G Fc Fragment Receptor and Transporter, has also been suggested to contribute to DM1 pathogenesis [87]. Indeed, Six5 knockout mice develop cataracts [88,89] but without any muscular deficiency. The next concept was the RNA gain-of-function hypothesis assuming that the mutant RNA transcribed from the expanded allele is capable of inducing symptoms of the disease. The HSALR transgenic mouse model (among others) confirms this theory [90]. In HSALR mice 250 CTG repeats were expressed in the 3′ end of the human skeletal α-actin gene that implied myotonia and muscle degeneration characteristic in DM1 without a multisystem phenotype.

DM2 was identified in 1998 with a different genetic mutation from that of DM1 [91]. In 2001, DM2 was reported as a result of CCTG repeats within intron 1 of the nucleic acid-binding protein (CNBP) gene (known also as zinc finger 9 gene, ZNF9) [92]. In both types of DM, there is a nucleotide repeat expansion; however, completely different genes are affected. Nevertheless, DM1 and DM2 have similar symptoms bringing up the idea of a common pathogenic mechanism. One candidate is a process through interaction with RNA-binding proteins. The transcripts with nucleotide repeat expansions can accumulate in the nucleus (see Figure 1) and form RNA aggregates/foci interfering with protein families such as the muscleblind-like (MBNL), CUGBP/Elav-like factors (CELF), and the RNA binding Fox (RBFOX) being the most important splicing regulators in skeletal muscle [93,94,95].

MBNL1 is sequestered on the expanded CUG repeats producing a loss of function, while CELF1 is upregulated due to the activation of protein kinase C, leading to its stabilization. These processes result in irregular splicing profiles of MBNL1- and CELF1-regulated transcripts in adult skeletal muscle and heart, or even during embryonic to adult switch in the splicing pattern. MBNL1 (Mbnl1ΔE3/ΔE3) knockout mice with targeted deletion of MBNL1 exon 3—where an RNA-binding motif is located—underpin this model, since these animals reproduce several features of DM including muscle, eye, and RNA splicing disorders (alternative splicing regulation in the brain is slightly affected, it depends mainly on the loss of MBNL2) [96]. Moreover, the adeno-associated virus-mediated overexpression of MBNL1 in HSALR mice is able to lessen the myotonia [24]. Verification also arises from the tissue-specific induction of CELF1 overexpression in adult mouse skeletal muscle, where muscle impairment detected in DM1 has been reproduced [97], while CELF1 overexpression in the heart leads to cardiac abnormalities similar to DM1 [98] (the role of CELF1 in DM2 is not clarified). These phenotypes were related to miss-splicing. These animal models imply that the overexpression of toxic CTG or CCTG repeats, depletion of MBNL1, or overexpression of CELF1 would all eventuate in similar splicing alterations, which initiate downstream signaling pathways resulting in the phenotype and/or molecular background of myotonic dystrophies.

The modified splicing apparatus can affect other genes in diverse signal transduction pathways leading to disrupted protein synthesis or the presence of different protein isoforms and the modified localization of proteins. Focusing on muscle, an aberrant regulation of RNA-binding proteins causes splicing alterations in the voltage-gated chloride channel 1 (CLCN1) transcript resulting in myotonia with delayed muscle relaxation in skeletal muscle cells [57,99]. Alternative splicing defects of BIN1 (bridging integrator 1), a lipid-binding protein responsible for the biogenesis of the transverse (T) tubules, has also been associated with muscle weakness. The inactive form of BIN1 causes damages of the excitation–contraction coupling (ECC) [100]. Another protein concerned is the calcium channel CaV1.1. Mis-splicing of the CACNA1S gene contributes to muscle weakness. Alternative splicing of the genes encoding ryanodine receptor 1 (RyR1) and SERCA1 expression are also altered modifying contractility of the muscles [101]. Altered splicing of several other genes encoding structural proteins has also been described in DM, as a few examples: DTNA (encoding dystrobrevin-α), MYOM1 (encoding myomesin1), NEB (encoding nebulin), TNNT3 (encoding fast troponin T3), DMD (encoding dystrophin), MTMR1 (encoding myotubularin-related protein 1), and CAPN3 (encoding intracellular protease Calpain 3). Atypical splicing of the insulin receptor (IR) may take part in the formation of insulin resistance. The genes mentioned above represent just a few examples of the more than 30 miss regulated splicing identified in DM patient’s tissue samples or of the more than 60 aberrant splicing described in mice tissues [102].

According to the previously listed conditions, molecular, and genetic variations, an “overall DM” model system should meet several requirements. At this moment, none of the available mouse models recapitulates all aspects of DM (for a comprehensive list of these mouse models, see Table 2). At the same time, the mouse models generated so far provided us with a significant tool in understanding the disease mechanism. There are different approaches in the various models. The inactivation of DM genes, the overexpression of toxic CTG/CTGG repeats, the induced alterations in splicing through MBNL1 inactivation, or CELF1 overexpression have resulted in a transgenic mouse model that was suitable for the examination of different aspects of the disease. Despite the growing number of already identified transcripts and the increased amount of data on altered pathways, the precise mechanism of the DMs is poorly understood. The available and the new mouse models to be established in the future can help scientists to discover a disease-modifying therapy.

### 2.3. Facioscapulohumeral Dystrophy

Facioscapulohumeral dystrophy (FSHD) also known as Landouzy–Dejerine syndrome is the 3rd most common autosomal dominant form of muscular dystrophy after DMD and DM. Its prevalence is 1:8500 to 15,000, and males are more often symptomatic compared to females [123]. The disease tends to progress slowly with periods of rapid deterioration, and it affects the face, shoulder blades, and upper arms muscles, leading to difficulty chewing or swallowing and slanted shoulders. Currently, there is no cure for FSHD, as no pharmaceuticals have proven effective for alleviating the disease course. Prognosis is variable, but most people with the disease have a normal lifespan. 

The FSHD is a very complex disease with primate-specific genetic and epigenetic components. It is caused by the epigenetic de-repression of the double homebox protein 4 (DUX4) retrogene on chromosome 4, in the 4q35 region that leads to a gain-of-function disease [124]. DUX4 is expressed in early human development, while in mature tissues, it is suppressed. In FSHD, DUX4 is inadequately turned off, which can be due to several different mutations. The mutation termed “D4Z4 contraction” defines the FSHD type 1 (FSHD1), making up 95% of all FSHD cases, whereas the disease caused by other mutations is classified as FSHD2 or contraction independent. 

There are a few FSHD1 mouse models available for preclinical efficiency testing prior to human clinical trials, but due to the unusual nature of the disease locus, these models will not recapitulate accurately the genetic and pathophysiological spectrum of the human condition, and overall, these models remain sub-optimal in assessing therapeutic efficacy (Table 3). The most significant hurdle that is impossible to overcome is that the D4Z4 macrosatellite encoding the toxic DUX4 retrogene is specific to primates, which impedes the possibility of working with a natural model of the disease [125]. Several xenograft models were developed in which skeletal muscle tissue from FSHD patients or muscle precursor cells were transplanted into the mouse muscle (see Table 3). There are currently no mouse models for FSHD2.

## 3. Myopathies

Myopathies contain a wide range of skeletal muscle disorders characterized by the irregular structure or muscle function. Myopathic patients show decreased physical activity without any disruption of sensory or autonomic function. Almost in all myopathies, symptoms affect proximal muscles bilaterally. Considering that a lot of myopathies cause progressive deterioration in the daily activity of patients, supportive therapy is often needed to overcome the physical and psychological effects of these diseases. Congenital myopathies are in a clinically, histopathologically, and genetically diverse group of rare hereditary skeletal muscle diseases that are characterized by structural abnormalities in the muscle fibers. They are subdivided into five subgroups: (1) congenital fiber-type disproportion myopathy; (2) centronuclear myopathies; (3) nemaline myopathies; (4) core myopathies; and (5) myosin storage (hyaline body) myopathy. The unusually broad genetic and clinical heterogeneity of these diseases stimulates for more expanded research on animal models. Furthermore, because of the lack of useful therapies, further studies are required to find candidates to cure patients with different types of myopathies.

### 3.1. Core Myophaties

Core myopathies are classified into congenital myopathies with variable clinical appearance, but they are usually associated with decreased muscle tone, pronounced muscle weakness and skeletal malformation; interestingly, symptoms do not or slowly progress with age.

#### 3.1.1. Multi Minicore Disease 

The neuromuscular illness multi-minicore disease (MmD) is characterized by multiple, amorphous cores seen on muscle biopsy and clinical features of a congenital myopathy. “Minicore” means that as a result of reduced or depleted oxidative activity, multiple core structures are visible in the muscle fiber [140,141]. MmD cores have a few or no mitochondria along with multiple internally placed nuclei, and type 1 fiber dominance is characteristic in the affected muscles (see Figure 1). 

Several forms of MmD have been identified. Among others, there are (1) the classic (responsible for ≈75% of all cases), (2) the progressive, (3) the antenatal, and (4) the ophthalmoplegic as the most common forms. The classic MmD has typical orthopedic disorders such as kyphoscoliosis [142] with respiratory abnormalities [143,144]. Genetic heterogeneity is responsible for the clinical variability. Recessive mutation of the ryanodine receptor gene can give a wide range of clinical features consisting of external ophthalmoplegia and distal weakness [145]. Recessive mutations in SELENON gene encoding selenoprotein N (SEPN1) [146,147] result in the classic phenotype, with spinal rigidity, respiratory impairment, and early scoliosis as typical characteristics. A severe form of MmD with bad prognosis can develop as a result of mutations in MYH7 gene encoding myosin heavy chain beta (MHC-β) isoform with cardiac involvement [148]. Mutations in MEGF10 encoding multiple epidermal growth factor-like domains protein 10 causes MmD with serious weakness, respiratory impairment, and scoliosis [149]. Mutations in CACNA1S (Voltage-Gated Calcium Channel Subunit Alpha1 S) or in SCN4A (Voltage-Gated Sodium Channel Alpha Subunit 4) have also been associated with MmD [150]. Mutations in TTN gene encoding the titin sarcomere component affect the Ig domain of the proximal I-band and can cause a congenital titinopathy, which manifests as an early onset of MmD without affecting the heart [151]. Rare MmD diseases with atypical cores caused by the autosomal-dominant CCDC78 (coiled-coil domain containing 78) mutations are diagnosed also as a centronuclear myopathy [152]. Last but not least, MmD can also be caused by mutations in ACTA1 [153], ACTN2 [154], and FXR1 [155], encouraging the creation of different non-ryanodine core myopathy mouse models to better understand these rare muscle disorders. As a result, various transgenic animal models have been developed to identify the disease progression mechanisms for some mutations in order to explain genotype–phenotype correlations. Without claiming completeness, the most studied core myopathy mutations in mouse models causing MmD are summarized in Table 4.

The most severe disorders of core myopathy are due to decreased RyR1 expression. In 1994, Takekura and colleagues developed a homozygous RyR1^-/-^ mice from the RyR1-knockout RyR1skrrm1 and RyR1tmAlle strain, in which the anchoring cytoplasmic ‘foot’ domain is missing. Homozygous RyR1^-/-^ mice die already perinatal due to respiratory failure and skeletal abnormalities. Interestingly, heterozygous RyR1skrrm1/+ and RyRtmAlle/+ mice do not have any apparent pathological irregularities [156,157]. This seems to correlate with human pathology; namely RyR1-associated core disorders caused by autosomal-dominant mutations or bi-allelic recessive RyR1 loss, rather than heterozygous loss [161]. 

In a recent study by Elbaz et al. (2019) a new mouse model was developed carrying heterozygous recessive RyR1 mutations isogenic with those identified in severely affected MmD patients (see Table 4). The authors came to the conclusion that the bi-allelic RyR1 p.A4329D mutation is responsible for a milder phenotype than its mono-allelic variant, causing changes in the biochemical properties and physiological functions, namely by focusing on the slowly twitch while sparing the fast twitch muscles [162].

As mentioned above, recessive mutations in SELENON gene encoding selenoprotein N (SEPN) result in an important part of the classical MmD. Since SEPN regulates Ca^2+^ levels in the ER/SR (sarcoplasmic reticulum) via SERCA2 activation [159], in Selenon1^-/-^ mice, myofibers have excessive oxidative/nitrosative stress and abnormal Ca^2+^ handling because of the dysfunctional ER-stress response and inhibited SERCA2 activity. Moreover, according to the experiments of Castets et al. (2011), in Selenon1^-/-^ mice, ER stress and high cytosolic Ca^2+^ levels caused impaired muscle regeneration deficiencies because of the reduced satellite cell numbers [163]. 

To the best of our knowledge, at the time of writing this review, there are no models for the extremely rare MYH7, for TTN and for ACTA1 mutations, which cause a subset of MmD with cardiac involvement [148,153,164].

#### 3.1.2. Central Core Disease

Central core disease (CCD) is a subgroup of core myopathies, an autosomal inherited muscle disorder, characterized by core-like lesions in myofibers [165,166]. RyR1 mutations are found in the background of the majority of cases (Figure 1); these missense substitutions mostly are identified in three hotspots: (1) in the N-terminal between C35 and R614, (2) central between D2129 and R2458, and (3) C-terminal regions between I3916 and G4942 in the amino acid sequence of RyR1 [167]. Several studies proved that CCD mutations enhance the sensitivity of RyR1, resulting in a gain of function. This alteration influences the ECC and Ca^2+^ homeostasis via leaky RyR1 channels and altered EC uncoupling mechanisms. In the case of CCD, either mechanism could explain why muscle weakness was observed in patients with the disease [68,168]. A decreased threshold for channel activation and impaired coupling between DHPR and RyR1 may be responsible for the observed pathological symptoms. 

Transgenic animals for CCD have been developed in order to study the altered protein function and pathological consequence. Malignant hyperthermia (MH) mutations in RyR1 might also cause CCD. The development of cores was formerly studied in two MH/CCD mouse lines: RyR1^Y522S/+^ and RyR1^R163C/+^. The Y522S and R163C mutations result in CCD in humans. Based on comprehensive studies, cores could not be detected in the RyR^1R163C/+^ mice [169,170]. However, RyR1^Y522S/+^ mouse showed progressive core development, and the localized regions containing damaged mitochondria were associated with disrupted sarcomeres and T-tubules, which could explain why cores were identified in patients with RyR1 mutations [171]. 

The RyR1^T4826I/T4826I^ knock-in mouse showed elevated resting Ca^2+^ levels; the mutant aged male mice had core myopathy-like features in the soleus muscle, including Z-line disorientation and impaired sarcomere organization [172]. Zvaritch and colleagues (2007) employed a knock-in mouse line expressing the EC-uncoupling RyR1 mutation, I4895T, which corresponds to one of the most common I4898T CCD mutations in humans, resulting in an EC-uncoupled phenotype due to the impaired function of RyR1. The heterozygous mutation causes severe clinical appearance in human patients. Ile-4898 is located in a highly conserved GGIG4899 motif, forming the selectivity filter of the Ca^2+^ release channel. Based on in vitro functional studies, all amino acid substitutions at position 4898 negatively influences Ca^2+^ release channel conductance. RyR1^I4895T/I4895T^ mice die perinatally because of paralyzed respiratory muscles [173]. Intact RyR1 Ca^2+^ release units and maintained SR Ca^2+^ content was detected in these mice; however, myofiber cultures showed disrupted RyR1-mediated Ca^2+^ release. Heterozygous RyR1^I4898T/+^ mice are born and despite exhibiting hypotonia and respiratory distress, they survive and do not show apparent skeletal deformities. On the other hand, mice have progressive congenital myopathy related to muscle weakness with age. These mice have been described to also develop cores, minicores, and rods [160,161]. The disease is slowly progressive, insufficient contractility was observed at the age of 2 months in fast and slow twitch muscles. Many mice show different degrees of impaired motor function at the age of 8 months. A combination of WT and mutant subunits randomly influences RyR1 functionality in a RyR1 tetramer that is proposed to contribute to phenotypic variability in RyR1-related disorders [161].

RyR1^I4898T/+^ mice (Table 4) are offered as the most appropriate genetically and phenotypically valid model of a RyR1-related congenital myopathy [160]. 

### 3.2. Centronuclear Myopathies

Centronuclear myopathy (CNM) is a general term for the family of rare genetic skeletal muscle diseases caused by a mutation in a definite gene. These disorders show muscle weakness ranging from mild to serious. Symptoms often start at birth in the serious forms of the myopathy, but they can also appear at any point during life, even though the onset in adulthood is rare. The name of CNM originates from the centrally located nucleus of the muscle fiber (normally located at the periphery). There are multiple genetic forms of CNM along with an X-linked form known as myotubular myopathy (XLMTM) caused by mutations in the myotubularin (MTM1) gene. There are a few autosomal forms as well, usually linked to three different genes: dynamin 2 (DNM2), bridging integrator 1 (BIN1), and RyR1 that have been identified to cause autosomal forms of CNM (Table 5).

#### 3.2.1. MTM1

X-linked myotubular myopathy (XLMTM) is a rare (1:50,000) congenital disease of skeletal muscle affecting only males [182]. XLMTM shares a general pathological trait in skeletal muscle, which is hypotrophic myofibers having centrally located nuclei [183,184,185]. The cause of the disease is mutations in the MTM1 gene encoding the universal phosphatase myotubularin, which plays a role in the phosphatidylinositol 3-kinase pathway to regulate intracellular vesicular transport and membrane trafficking [186,187,188,189,190]. MTM1 has effects on both types of phosphatidylinositol–phosphate [PtdIns(3,5)P2 or PtdIns(3)P] [186,191,192,193]. To date, more than 200 loss-of-function mutations of the MTM1 gene have been found in myotubular myopathy patients [194,195,196].

XLMTM patients are classified into three groups (mild, intermediate, or severe phenotype) based on remaining ventilator capacity [197]. Most of the patients have the severe phenotype showing serious hypotonia and a lack of spontaneous breathing at birth [198,199]. These newborns usually die within the first months of life. Long-term survivors in this group need continuous ventilation support [198,200]. Patients with mild and intermediate phenotypes can breathe independently at least a few hours daily [197,200].

Studies performed on mouse models lacking myotubularin have shown that skeletal muscle is the prime tissue affected in the pathogenesis of myotubular myopathy, and the protein is necessary for proper muscle development and the normal distribution of myofibrillar organelles. MTM1 knockout (KO) mice evolve centronuclear myopathy, starting at around one month after birth, showing dynamic muscle weakness that critically decreases lifespan to a maximum of 2–3 months [174].

Al Qusairi and colleagues reported that ECC of skeletal muscle is the major target of myotubular myopathy [201]. Using MTM1 KO mice, they showed that myopathic muscle fibers have abnormal longitudinally oriented T-tubules and a decreased number of triads (the structure formed by a T tubule with a sarcoplasmic reticulum (SR) on both sides; Figure 1). As a consequence, depolarizations evoked a calcium release from the SR that is strongly decreased while the SR calcium content and the removal of Ca^2+^ from the myoplasm were unaffected. These changes were accompanied with the 3-fold reduction in the level of RyR1. The authors hypothesized that the abnormal SR Ca^2+^ release causes the failure of muscle function in MTM1 KO mice.

Further investigation of MTM1 KO mice revealed the fact that the blockade of phosphatidylinositol 3-kinase (PtdIns 3-kinase) activity restores the defected Ca^2+^ release from the SR in isolated muscle fibers and increases the mobility and extends the lifespan of these mice [202]. The same group showed that the muscle fibers of MTM1 KO mice exhibit spontaneous elementary Ca^2+^ release events (sparks) with 30 times higher frequency than control fibers at resting conditions. These sparks occur at locations in the fibers where RyR1s lack the control of the voltage sensor DHPR because of the disrupted T-tubule membrane [203].

In the meantime, another murine model of XLMTM was developed by introducing a c.205C > T base change in MTM1 exon 4 [175]. These mice have a longer lifespan than MTM1 KO mice and show a milder MTM phenotype; however, they still present significant muscle weakness and atrophy. The genetic defect of these mice has a human counterpart. 

To date, no effective treatment exists for XLMTM patients; however, other potentially usable therapeutic targets were suggested according to animal studies. The same authors who developed the MTM1 KO mouse proved that one intramuscular inoculation of myotubularin expressing adeno-associated virus (AAV) in MTM1 KO mice reverted the pathological phenotype in the injected muscle. The myotubularin replacement substantially corrected mitochondria and nuclei positioning in myofibers. These positive changes greatly increased muscle volume and force [204]. Later, they suggested enzyme replacement therapy, since myotubularin is a cytoplasmic enzyme, it does not have mannosylation, and it circulates in the blood. By delivering myotubularin in a fusion protein form, researchers were able to improve the structure and function of MTM1 KO muscle [205]. 

Dowling and colleagues showed that abnormal neuromuscular junction (NMJ) signal transmission is a crucial and likely manageable aspect of the MTM1 disease pathogenesis. An acetylcholinesterase inhibitor treatment significantly improved the fatigability and treadmill performance in MTM1 KO mice [206]. However, this type of treatment has not been tried in human therapy yet.

Inhibition of the phosphoinositide 3-kinase PIK3C2B improved the motor function and prolonged lifespan of the MTM1-deficient mice [207]. Another study on double KO mice demonstrated that the reduction of dynamin 2 (DNM2) expression in MTM1^–/y^ mice was enough to decrease the early XLMTM lethality as well as most hallmarks of the disease; it also increased the lifespan of mice [176]. A systemic application of DNM2 antisense oligonucleotides in MTM1-KO mice was shown to prevent the development of muscle myopathy by reducing the DNM2 protein level [208]. In addition, this type of treatment in severely affected mice reversed the muscle pathology within 2 weeks.

Recently, two studies presented beneficial effects of long-term tamoxifen treatment, which increased the lifespan of MTM1 KO mice by improving the overall motor function [209,210]. Tamoxifen, a selective estrogen receptor modulator used in breast cancer therapy, eliminated successfully the molecular, histological, and functional hallmarks of XLMTM. Tamoxifen is the first long-term used and safe drug with a promising therapeutic potential for XLMTM patients.

A fresh study introduced a miR-199a-1-MTM1 dko mice model [177]. This research group demonstrated an upregulation in the expression of the intragenic microRNA miR-199a-1 and DNM2 as a host gene in XLCNM skeletal muscle. The dko mice displayed longer lifespans and improved muscle histology and strength. Their results suggest that this microRNA is a potential target in therapies to manage XLCNM.

#### 3.2.2. Dynamin2

DNM2-related myopathies are the consequence of a missense mutation in the dynamin 2 (DNM2) gene, leading to an autosomal congenital dominant disease. Some cases of DNM2-related CNM may occur spontaneously (sporadically) with no previous family history of the disorder (i.e., new mutations) [211]. The encoded protein is universally expressed and associated to membrane trafficking and endocytosis, and it plays a role in centrosome cohesion and actin assembly. This large GTPase protein has five functionally distinguishable domains: the N-terminal domain is the GTPase; the middle domain (MD); the domain homologues to pleckstrin (PH); the GTPase effector domain; and it ends in an arginine and proline-rich domain at the C-terminal (PRD) [212]. The foremost found patients suffering in DNM2-related autosomal dominant CNM showed a slowly progressing muscle weakness, and the disease affected mainly distal muscles with onset in early adulthood. Shortly afterwards, four new mutations in the DNM2 gene were found in children presenting neonatal hypotonia manifested in weak suckling as well as lower limb and facial muscle weakness [213]. The number of DNM2 mutations increased when new missense mutations in the PH domain in the C-terminal region were found [36]. These genetic failures were associated with a very severe clinical phenotype present from infancy but also in adults. To date, a little over more than 100 human mutations have already been reported in DNM2 gene with different onset and phenotypes [214], and from these, 35 human mutations of the DNM2 gene have been identified associated to CNM. The only common characteristic is the morphological hallmarks: hypotrophic fibers with centralized nuclei. In the past few years, several new human mutations of dynamin2 were identified (i.e., p.G359D in the middle domain by Chen et al., 2018). It should be noted that patients with mutations of DNM2 often present a disorder of the peripheral nerve (Charcot–Marie–Tooth disease).

The first murine model of DNM2 mutation was generated by Durieux and colleagues (2010, KI-DNM2^R465W^). This knock-in (KI) heterozygous mouse model mimicked the most common mutation in the DNM2 human gene known at that time. KI mice showed progressively developing muscle weakness from 3 weeks of age, and atrophy developed at around 2 months of age. The membrane trafficking was severely altered, and a high level structural disorganization of muscle fibers (Figure 1) was accounted as the main mechanism of the disease [214]. A modified intracellular Ca^2+^ homeostasis was also reported: the resting intracellular [Ca^2+^], the sarcolemmal calcium permeability, and the releasable SR Ca^2+^ content was increased in muscle fibers from KI-DNM2^R465W^ mice [215,216]. This mouse model was further investigated in detail, and the density of the calcium current through DHPRs and the rate of voltage-activated SR calcium release were found to be reduced. Fibers from the mutant KI mice produced elusive spontaneous Ca^2+^ release events under resting condition, which were not present in control animals [217].

Another research group developed a double KO mouse model lacking miR-133a-1 and miR-133a-2 showing progressive CNM [179]. The myopathy was accompanied with mitochondrial dysfunction, which can be attributed partly due to the upregulation of DNM2 [179]. These mice showed T-tubule disorganization, leading to impaired EC coupling functions.

As previously mentioned, the reduction of DNM2 expression improved the lifespan of XLMTM animals (see the subsection on MTM1) [176]. The same research group proved also that the DNM2 modulation can be used as a therapeutic application for patients with BIN1 defects. BIN1-related CNM is caused by mutations to the amphiphysin 2 (BIN1) genes and is inherited as an autosomal recessive condition. BIN1 and DNM2 are ubiquitous proteins involved in membrane remodeling. Cowling and colleagues generated BIN1 and DNM2 double KO mice that survived at least one and half years and had maintained muscle force and a normally organized structure of muscle fibers. The authors have hypothesized that DNM2 and BIN1 regulate muscle maturation and work through a common pathway, and they depicted BIN1 as negatively regulating DNM2. It was shown that lowering the level of DNM2 after birth could be sufficient to turn back the decline of muscle functions and progression of XLMTM [180].

If we take into account the fact that the total elimination of DNM2 is lethal at embryonic stages in mice, but the heterozygous KO mice are viable with unaffected muscle function, a potential therapeutic approach can be the reduction of the expression of the mutant allele without affecting the wild-type allele [176]. This concept was established by developing allele-specific siRNA sequences to specifically reduce the human and murine DNM2–mRNA containing the p.R465W mutation [218]. The technique resulted in a promising functional restoration of muscle function in mice.

Following the same train of thought, another research group used a single intramuscular injection of adeno-associated virus-shRNA against DNM2 in a knock-in mouse harboring the p.R465W mutation. Five weeks post injection, the fiber size distribution and muscle mass were improved [219]. The authors established also a systemic treatment by using intraperitoneal injections of antisense oligonucleotides against DNM2 weekly for 5 weeks. This treatment was similarly successful in minimizing pathological symptoms in DNM2^R465W/+^ mice.

CRISPR/Cas9 technology for genome editing is a recently emerging elegant technique. This was used in a study investigating an allele-specific correction or inactivation of a heterozygous mutation in the DNM2 gene. DNM2^R465W/+^ murine myoblasts showed less hallmarks of the disease after CRISPR/Cas9 correction of the dominant point mutation [220].

A fresh study targeting the S619L missense mutation successfully used DNM2 reduction with antisense oligonucleotides. Histological, force, and locomotor defects were partially or fully rescued just after 3 weeks of treatment in mice [181].

## 4. Malignant Hyperthermia

One of the most severe emergency situations that may occur in the operating room is caused by malignant hyperthermia (MH) susceptibility (MHS) of the patient. MH syndrome is an idiosyncratic reaction to volatile anesthetics such as halothane, isoflurane, desflurane, sevoflurane, and the depolarizing muscle relaxant succinylcholine. Symptoms include general muscle contracture, which leads to a rapid increase of the body temperature (1 °C/5 min), lactic acidosis, and hyperkalemia. These symptoms are likely to be fatal unless the patient is immediately treated with the muscle relaxant dantrolene and the body is cooled down [221,222]. 

The prevalence of MH crisis ranges from 1:5000 to 1:50,000 anesthesia. In the past 40 years—since dantrolene sodium must be available in all operating rooms—the mortality of MH dropped from over 80% to less than 5% [223,224]. Fortunately, animal models since the 1970s made great progress in understanding the pathophysiology and clinical manifestation of MH. The most widely used experimental animals were pigs from certain pig breeds, such as Pietrain, Landrace, Yorkshire, and Poland China, which were affected by MHS. The genetical cause that accounts for the syndrome was discovered by MacLennon’s group, who identified a common mutation (R615C) in the gene encoding the skeletal muscle type RyR (Figure 1), which was responsible for the porcine MHS phenotype, suggesting that a RyR1 mutation is linked to human MHS, too [225,226]. Since then, more than 200 MHS mutations have been identified in the human gene [227]. They are clustered in three mutation hotspots (N-terminal, aa 35–614; central, aa 2163–2458; C-terminal, aa 4550–4940) [228]. Mutations are believed to destabilize the closed-state conformation of the Ca^2+^ release channel (i.e., RyR1); therefore, all these MH-susceptible RyRs share a common, overactive, gain-of-function phenotype [229]. Hypersensitive gating has been demonstrated in response to major RyR agonists (such as caffeine and ATP), and most importantly for Ca^2+^ [230,231]. This feature creates a low, unsafe stimulation threshold for halothane, resulting in uncontrolled Ca^2+^ release and a consequent contracture of resting muscles when exposed to therapeutic concentrations of volatile anesthetics. In addition, diminished inhibition by Mg^2+^ has also been demonstrated, which may contribute to the pathogenesis, too [232,233]. Although the porcine model was extremely useful in preclinical studies of the RyR inhibitor dantrolene [233,234], this model has many disadvantages [235,236,237,238,239,240,241]. For example, R615C is a recessive mutation in pigs. Moreover, R615C represents only 2% of all human mutations. Apparently, the detailed understanding of the pathomechanism of MH required genetically modified mouse models covering all three hotspots. To date, four MHS RyR1 knock-in muse genotypes are available: Y524S, R163C, G2435R, and T4826I [169,172,230,242,243,244,245]. All these mice reproduce a typical MHS phenotype, displaying whole body contractions and elevated core temperatures in response to therapeutic concentrations of halothane or isoflurane. Similar to pigs, MHS mice exhibit heat-stress-induced MH episodes. Common features of muscle fibers or myotubes include elevated resting intracellular Ca^2+^ concentration and increased susceptibility to caffeine- and heat-induced contractures in vitro.

Y524S was the first murine model of human MHS (Y522S). Homozygous mice show severe skeletal and muscular abnormalities and die at the early stage of intrauterine life (17th day) or soon after birth. Heterozygous mice are viable and reproductive [169,242,243]. R163C (also R163C in human) is a dominant heterozygous mutation with no phenotype until exposed to a trigger agent. Homozygous are not viable at birth [230,244]. The mouse carrying the mutation G2435R is the model for the most common human MHS mutation G2434R. G2434R mutation has been found in 16% of families tested. Both homozygous and heterozygous mice are viable and fertile, although some homozygous males died spontaneously [245].

KI mice heterozygous or homozygous for T4826I RyR both survive, although homozygous animals were more sensitive to halothane and heat stress. In addition, males were reported to be more susceptible to MH trigger agents than females [172].

In summary, all four MHS RyR KI mouse strains summarized in Table 6 accurately mimic the patient’s phenotype and provide invaluable tools to investigate the detailed pathomechanism of MH and will be useful in the future to discover new, potential trigger agents [246,247].

## 5. Conclusions

Although skeletal muscle disorders represent rare diseases and affect only a smaller portion of the population, the disability imposed on the affected person as well as the necessity to care for these individuals carries a significant economic burden for the society and the healthcare system. This justifies the need for appropriate animal models in developing new therapies and testing the proposed interventions to understand the nature of the given disorders.

The present review summarizes the most commonly used mouse models for a subset of muscle disorders with the highest prevalence in the human population; these mouse models provide important insights into causal gene relationships, have forged our understanding of molecular mechanisms and disease pathogenesis, and have driven progress toward a cure for muscle disorders. Most of the skeletal human diseases have the afferent mouse models, whose human relevance is still pending as these models are limited in their presentation of the human pathologies; however, there are promising results based on the recent advances achieved via elegant approaches such as gene editing or alternative splicing. Unfortunately, only a fraction of muscle disorders possess effective therapy at this moment, so finding and/or establishing an appropriate animal model is an important step toward the direction of understanding the complex pathomechanism of the disease and ultimately develop effective therapies.

For the existing mouse models summarized in Table 1, Table 2, Table 3, Table 4, Table 5 and Table 6, in most cases, therapies have been tested, although some of these have been not implemented in human treatment; nevertheless. They represent promising approaches that should eventually reach clinical trials. Yet, one has to acknowledge the obvious differences between the two species as well as the difficulties in targeting genetic therapies in human patients.

Taken altogether, we hope that this summary will help and encourage the scientific community to continue the search for proper animal models and therapies for muscle disorders.

## Figures and Tables

**Figure 1 ijms-21-08935-f001:**
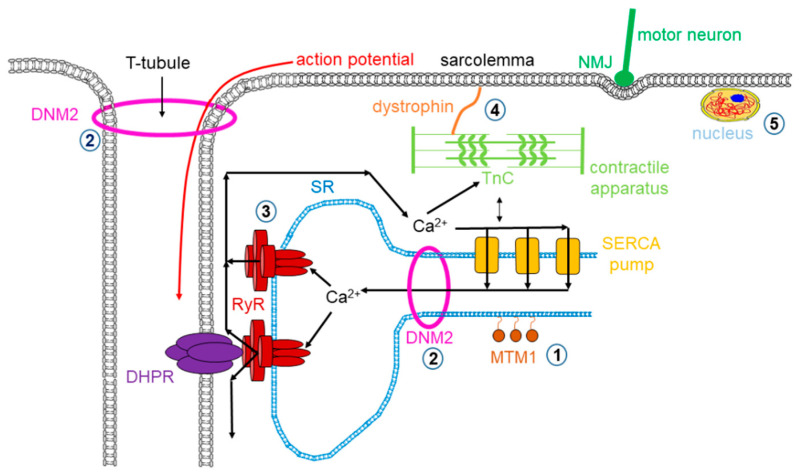
Mutations in skeletal muscle causing muscle disorders. **➀**. Altered lipid phosphatase activity of MTM1; **➁**. defects in microtubule dynamics or vesicular traffic (DNM2); **➂**. defective calcium release from the SR via RyRs; **➃**. mutated or missing dystrophin; **➄**. defects in alternative splicing due to MBLN1, CELF1 and DUX4 malfunction. (MTM1: myotubularin 1; DNM2: dynamin 2; SR: sarcoplasmic reticulum; RyR: ryanodine receptor; MBLN1: muscle blind-like; CELF: CUGBP/Elav-like factors; DUX4: homebox protein 4; DHPR: dihydropyridine receptor; NMJ: neuromuscular junction; SERCA: sarco(endo)plasmic reticulum calcium pump).

**Table 1 ijms-21-08935-t001:** Mouse models of Duchenne muscular dystrophy.

Model System	Genetic Changes in the Mouse Model/Mutation(s)	Genetic Similarity/Genetic Background		Likeliness of Phenotype and Symptoms		References
		Advantage	Disadvantage	Advantage	Disadvantage	
**Dystrophin-deficient mice**						
mdx Albino mdx mdx/BALB/c mdx/BL6 mdx/C3H mdx/DBA2 mdx/FVB	Exon 23 point mutation.	On the C57BL/10 background. mdx on the Albino background. mdx on the BALB/c background. mdx on the C57BL/6 background. mdx on the C3H background. mdx on the DBA2 background. mdx on the FVB background.		The diaphragm shows progressive deterioration as in humans. More severe dystrophic signs.	Minimal clinical symptoms, lifespan reduced by only 25% compared to human DMD.	[15,16,17,18,19]
mdx^2cv^ mdx^3cv^ mdx^4cv^ mdx^5cv^	Intron 42 point mutation. Intron 65 point mutation. Exon 53 point mutation. Exon 10 point mutation.	On the C57BL/6 background.	Chemically induced mutation. all dystrophin isoforms eliminated.	Fewer revertant fibers. Severe disease signs.		[20]
CRKHR1	Unsequenced, dystrophin deficiency confirmed by immunofluorescence staining.	On the C3H background.	ENU chemically induced mutation.	Elevated CK, centrally nucleated myofibers, and dystrophin deficiency.		[21]
mdx52	Exon 52 deletion.	On the C57BL/6 background hot spot mutation.	Targeted inactivation.			[22]
mdx^βgeo^	Insertion of the β-geo gene trap cassette in intron 63.	LacZ replaced the CR and CT domain.	All dystrophin isoforms are mutated.			[23]
DMD-null	Entire DMD gene deletion.	Cre-loxP system.	All dystrophin isoforms are eliminated.			[24]
Dp71-null	Insertion of the β-geo cassette in intron 62.		Selective elimination of Dp71.	Dp71 deficiency is associated with early cataract formation in mice.		[25,26]
Dup2	Exon 52 duplication.	On the C57BL/6 background.				[27]
**Immun-deficient mdx mice**						
NSG-mdx^4cv^	Prkdc and IL2rb double deficient.	On the mdx4cv background.	Innate immunity deficient.	B, T, and NK cell deficient.		[28]
Rag2 IL2rb Dmd	Rag2 and IL2rb double deficient.	On the mdx βgeo background.		B, T, and NK cell deficient. No revertant fibers.		[29,30]
Scid mdx	Prkdc deficient.	On the mdx background.		B and T cell deficient.		[31]
W41 mdx	C-kit receptor deficient	On the mdx background	Haematopoietic deficient.	Optimal for bone marrow cell therapy studies.		[32]
**Phenotypic dko mice**						
α7/dystrophin dko or mdx/α7^–/–^	α7/dystrophin double deficient.			Severe dystrophic phenotype.		[33,34]
Adbn^–/–^ mdx	αdystrobrevin/dystrophin double deficient.			Severe dystrophic phenotype.		[35]
Cmah-mdx	Cmah/dystrophin double deficient.			Severe dystrophic phenotype. “humanized”		[36]
d-dko	δ-sarcoglycan/dystrophin double deficient.			Severe dystrophic phenotype.		[37]
Desmin^-/-^ mdx4cv	desmin/dystrophin double deficient.			Severe dystrophic phenotype.		[38]
Dmd^mdx^/Large^myd^	like-glycosyltransferase/dystrophin deficient.			Severe dystrophic phenotype.		[39]
DMD null; Adam8^-/-^	ADAM8 deficient and entire DMD gene deletion.	On the DMD-null background.	The injured myofibers are not efficiently removed in DMD null.			[40]
dysferlin/dystrophin dko	dysferlin/dystrophin double deficient.			Severe dystrophic phenotype.		[41,42]
Il-10^-/-^/mdx	interleukin-10/dystrophin double deficient.	On the mdx background.		Severe dystrophic phenotype and marked cardiomyopathy.		[43]
mdx/mTR	telomerase RNA/dystrophin double deficient.		Premature depletion of myofiber repair.	Severe dystrophic phenotype.		[44]
mdx:MyoD^-/-^	MyoD/dystrophin double deficient.		MyoD is only expressed in skeletal muscle.	Severe dystrophic phenotype and prominent dilated cardiomyopathy.	MyoD mutations do not occur in human DMD.	[45]
mdx:utrophin^-/-^	utrophin/dystrophin double deficient.		Targeted mutation at the utrophin CR domain/exon 7.	Severe dystrophic phenotype with cardiomyopathy, cardiac fibrosis, LV dilation.		[35,46]
PAI-1^-/-^-mdx	plasminogen activator inhibitor-1/dystrophin double deficient.		Early onset fibrosis and higher CK.			[47]
**Transgenic mdx mice**						
full-length dystrophin TG mdx	transgenic over-expression of full-length dystrophin.	On the mdx background.		Over-expression does not harm muscle rather it shows protection.		[48,49,50]
Dp71 TG mdx	transgenic over-expression of Dp71.	On the mdx background.		Severe disease signs.		[51,52]
Dp116 TG mdx4cv Dp116:mdx:utrophin^-/-^	transgenic over-expression of Dp116.	On the mdx4cv background on the utrophin/dystrophin dko background.		Severe disease signs. Improved lifespan.	No change in histopathology, CK, and specific force development.	[53,54]
Dp260 TG mdx Dp260 mdx/utrn^-/-^	transgenic over-expression of Dp260.	On the mdx background on the utrophin/dystrophin dko background.		Slightly improved histopathology. Severe lethal phenotype was converted to a mild myopathy.	No improvement of specific force.	[55,56]
micro-dystrophin TG	transgenic over-expression of synthetic micro-dystrophin gene.	On the mdx background.		Improved protection against disease signs.	No restoration of nNOS.	[57,58,59,60]
Fiona	transgenic over-expression of full-length dystrophin gene.	On the mdx background.		Improved protection against disease signs.	No restoration of nNOS.	[61,62]
laminin α1 TG mdx	transgenic over-expression of laminin α1.	On the mdx background.		Similar phenotype as mdx.	No improvement but no harm.	[63]

**Table 2 ijms-21-08935-t002:** Mouse models of myotonic dystrophies.

Model System	Genetic Changes in the Mouse Model/Mutation(s)	Genetic Similarities/Genetic Background		Likeliness of Phenotype and Symptoms		References
		Advantage	Disadvantage	Advantage	Disadvantage	
DMPK KO	Reduced DMPK transcripts levels by inactivation of the DMPK gene.	Can be used to study relief pathways in DM pathogenesis. Lacks RNA toxicity and transcripts interactions.			Increased possibility of cataracts, male infertility, and cardiac dysfunction. No characteristic symptoms on different organs.	[84,85,88,89,103,104,105]
Tg26	Overexpression of normal DMPK gene with short, non-pathogenic CTG repeats.	Can be used to study the effect of normal DMPK in high expression levels.	The pathogenesis is vastly different from conventional DM.	Severe cardiomyopathy symptoms, skeletal muscle wasting, and smooth muscle weakening.	Lack of non-muscle-like symptoms.	[85,106]
HSA^LR^	High levels of skeletal muscle expression of untranslated CUG repeats (≈250) in an unrelated mRNA.	The effect of CUG repeats in RNA and nuclear foci can be studied.	Interaction with transcription factors may be different from conventional DM.	High lethality in early developmental stages, myotonic discharges in young animals, myopaty in later stages.	Lack of muscle wasting and other neurological effects; the NMJ cannot be studied in depth.	[90,107]
DMSXL	Expanded DMPK transcript expression with different repeat sizes in various mouse tissues driven by *cis*-regulated human DM1 locus fragment.	Accumulation of ribonuclear foci and abnormal splicing patterns in multiple tissues in homozygous DM300.	Possible dose-dependent RNA toxicity. Time-consuming and costly mouse breeding. The correlation of copy number and phenotype is hard to quantify.	Skeletal muscle, cardiac and CNS symptoms such as myotonia, progressive muscle weakness, age-dependent glucose intolerance.	Relatively lower expression levels of the CUG-containing transcripts compared to other mouse model systems that lead to milder symptoms.	[108,109,110,111]
EpA960	Cre-loxP system induced tissue-specific expression of DMPK exon 15 with large iterrupted CTG repeats.	Transcripts foci accumulation, MBNL1 sequestration, CELF1 upregulation, and the return of embryonic splicing patterns.	Due to tissue specificity, the complex multisystem symptoms of DM are hard to model but with leaky EpA960 transgene expression is manageable.	In cardiac tissue severe histopathological, functional and electrophysiological changes. In skeletal muscle, the Cre-loxP system induced myotonia and muscle weakness with progressive status.	Due to tissue specificity, the complex multisystem symptoms of DM are hard to model.	[112,113]
GFP-DMPK-(CTG)_X_	Expression of the DMPK 3′UTR with different repeat sizes.	The extent of RNA toxicity shows CUG-triplet repeat dose effect on myogenesis in overexpressing model of DMPK 3′UTR, which can be compared in distinct repeat expansions.	The expression rate and the length of CUG repeat can affect the pathomechanism of DM differently.	In higher repeat numbers, the DM phenotype was present, and increased CUG expansion amplified the symptoms.	With small repeat numbers, the model failed to produce skeletal muscle atrophy, due to premature death caused by severe cardiac damage.	[111,114]
Mouse line to model abnormal splicing regulators connecting DM	Modeling MBNL sequestration by KO or propagating alternate splicing patterns by overexpressing CELF.	Simulation of downstream changes of DM by knocking out MBNL or overexpressing CELF.	The interactions of the protein family MBNL show a combinatorial loss-of-function nature and with the different expression levels of CELF, the system may show high variability.	Typical DM symptoms in various tissues: cataracts, motivation deficits and apathy, cardiac conduction defects.	Muscle weakness or muscle waisting was not detected. Histological, functional, and molecular changes were based on the rate of CELF upregulation.	[115,116,117,118]
DSMD-Q KO	Loss of function variants (frameshift, insertion, or deletion) induced by CRISPR-Cas9 to Dmpk, Six5, Mbnl1 and Dmwd genes.	Combines the three approaches of DM1: the haploinsufficiency model, the RNA toxicity model, and the chromatin structure malformation model.	Off-target problems of the CRISPR-Cas9 method dismissed by whole genom sequencing.	Conventional DM1 symptom: skeletal muscle wasting and weakness with correlating histopathology; heart problems; endocrine disorders; pathological changes in the digestive tract and neurological impairment caused by satellite cell malfunction.	Can simulate the characteristics of DM1 but not suitable for DM2.	[119]
Mouse lines to model downstream components of DM: Cav1.1e CLCN1 BIN1 Insulin receptor	Alternative splicing variants of ion channels and/or receptors lead to the expression of embryonic form of channels and/or mutated receptors through development.	The effect of ion channels and/or metabolic pathway receptor misplicing can be studied separately from other genetical changes.	The genetic background vastly different from the conventional DM model lines such as RNA toxicity or haploinsufficieny approaches.	Can be used to distinguish the role of downstream components of DM pathomechanism.	CaV1.1 mainly affects intracellular calcium homeostatis such as mitochondria but not linked closely to other aspects of DM. CLCN1 mainly affects the conductive properties of excitable cells.	[57,120,121,122]

**Table 3 ijms-21-08935-t003:** Mouse models of facioscapulohumeral muscular dystrophy.

Model System	Genetic Changes in the Mouse Model/Mutation(s)	Genetic Similarities/Genetic Background		Likeliness of Phenotype and Symptoms		References
		Advantage	Disadvantage	Advantage	Disadvantage	
AAV6-DUX4	TA injection of AAV6-DUX4 in 6–8-week-old mice.	On the C57BL/6 background.		Degenerating myofibers and infiltrating mononuclear cells. DUX4-induced cell death via p53-dependent pathway.	Minor degeneration, increased central nuclei. Signs of apoptosis.	[126]
D4Z4-2.5 D4Z4-12.5	Transgenic insertion of two and a half copies of D4Z4 from the permissive haplotype of a pathogenic allele. Transgenic insertion of twelve and a half copies of D4Z4 from the permissive haplotype of a pathogenic allele.	On the C57BL/6NJ background. Body-wide expression of the DUX-4 transcript in all tissues.	Keratitis leading to blindness. DUX4 transcript detected in myoblasts and myotubes. DUX4 transcript was NOT detected in tibialis anterior and pectoralis muscles.	No muscle weakness or abnormal morphology.	Satellite-cell derived myoblasts with DUX4 positive nuclei fail to fuse and form myotubes. Minor regeneration defect upon cardiotoxin injury.	[127]
iDUX-2.7 iDUX4pA	Doxycycline-inducible DUX4 transgene on the X-chromosome.	On C57BL/6J background.	Abnormal embryogenesis, mostly lethal. Surviving males lived ‹ 2 months.	Weaker grip strength. Smaller muscles, impaired function, reduced specific force.	Impaired myogenic regeneration. The activation of the downstream targets of DUX4 in mice differs from that in humans. Smaller and fewer myofibers, but not dystrophic. TA was least affected.	[128,129]
Xenograft	Human muscle engraftment into immunodeficient mice.	“Humanized” mouse model.		FSHD biomarker profile maintained in xenograft.		[130,131,132]
FRG1	Transgenic insertion of FRG1 driven by a human skeletal α-actin promoter.	Spinal curvature correlated with the level of FRG1 expression. Dystrophic features.		Fiber size variability, necrosis, centralized nuclei. Excess collagen, selective muscle atrophy, reduced exercise tolerance.	Abberant alternative splicing of specific pre-mRNAs.	[133]
Fat1	Knockout of Fat1.	Regionalized muscle and non-muscle abnormalities.	Retinal vasculopathy, abnormal inner ear patterning. Abnormal embryogenesis.	Muscle weakness of the face and scapulohumeral region.	Altered myoblast migration polarity.	[134]
Pitx1	Transgenic overexpression of Pitx1 induced in the absence of doxycycline.	Myofiber atrophy, necrotic and centrally nucleated fibers, inflamatory infiltration.	Polyadenylated DUX4 mRNA expressed at higher level in FSHD muscle.	Asymmetric muscle weakness in the face and shoulders that gradually progresses into the trunk and leg muscles. Evidence of endomysial inflamation.	Retinal vasculopathy hearing loss.	[135,136,137]
TIC-DUX4 FLExDUX4	Tamoxifen inducible Cre-DUX4	Reproductively viable.	No functional deficit of diaphragm muscles. Progressive pathology. Mild alopecia. Females more affected.	AAV-mediated follistatin gene therapy improved muscle mass and strength. No extramuscular deficits.	Tamoxifen dose-dependent skeletal muscle pathology. Limited skeletal muscle pathology.	[138,139]

**Table 4 ijms-21-08935-t004:** Mouse models for multi-minicore disease and central core disease.

Model System	Genetic Changes in the Mouse Model/Mutation	Genetic Similarity/Genetic Background		Likeliness of Phenotype and Symptoms		References
		Advantage	Disadvantage	Advantage	Disadvantage	
Homozygous RyR1^-/-^ mice	From RyR1^skrrm1^ (RyR1-knockout) and RyR1^tmAlle^ (foot domain is missing) strains.		RyR1-associated core disease is caused by autosomal-dominant mutations or biallelic recessive RyR1 loss.		The mice die perinatal as a result of respiratory failure due to a lack of ECC, severely reduced muscle mass, skeletal abnormalities.	[156,157]
Heterozygous recessive RyR1 mice	Frameshift RyR1 p.Q1970fsX16 mutation in exon 36 plus the missense mutation RyR1 p.A4329D in exon 91.	Isogenic with those identified in severely affected MmD patients.		The bi-allelic RyR1 p.A4329D mutation causes a milder phenotype than its monoallelic expression.		[156,157,158]
Selenon1^-/-^	Sepn1 <tm1.2Mred>/Orl.	SEPN1 KO mice are protected from the effects of SEPN1 loss.	Unclear why they do not show muscle phenotype.	Dysfunctional ER-stress response and inhibited SERCA2 activity; depleted mitochondria, minicores.		[159]
**Central Core Disease**						
RyR1-related congenital myopathy.	Missense substitution I4895T. Isoleucine–threonine.	Genetically and phenotypically valid model of a RyR1-related congenital myopathy.	Similar pathogenic phenotypes can arise from functionally different RyR1 mutations.	Progressive congenital myopathy related to muscle weakness with age. Mice also develop cores, minicores, and rods.	Phenotypic variability in RyR1 functionality.	[160]

**Table 5 ijms-21-08935-t005:** Mouse models for centronuclear myopathies.

Model System	Genetic Changes in the Mouse Model/Mutation	Genetic Similarity/Genetic Background		Likeliness of Phenotype and Symptoms		References
		Advantage	Disadvantage	Advantage	Disadvantage	
MTM1δ4 (MTM1^−/y^) MTM1-deficient	Absence of exon 4 in MTM1.	Total loss of myotubularin.	Female mice are also affected.	Very short lifespan, accumulation of central nuclei in skeletal muscle fibers, progressive and generalized myopathy starting at around 1 month of age.	Humans have different clinical evolution of the disease. XLMTM patients show severe myopathy at birth, which appears to be non-progressive.	[174]
MTM1 p.R69C	c.205C>T base change in MTM1 exon 4.	Exon 4 skipping in the mouse similar to human MTM, as in quadriceps from a patient with the c.205C>T mutation.	There are some other human mutations: c.C208T (p.L70F), c.C205A (p.R69S), c.T202G (p.Y68D).	Longer lifespan than MTM1 KO mice and milder MTM phenotype with significant muscle weakness and atrophy.	Some residual myotubularin activity remains in MTM1 p.R69C mice.	[175]
MTM1^–/y^ DNM2^+/–^	MTM1^–/y^ mouse that is heterozygous for DNM2.	50% reduction of DNM2.		Longer lifespan than MTM1^–/y^ mice. Similar to WT.		[176]
MTM1^Δ5/y^, MTM1^Δ7/y^	5-bp (MTM1^Δ5/y^) and 7-bp (MTM1^Δ7/y^) deletion within the MTM1 gene with CRISPR-Cas9 technology.			Similar genotype as MTM1^−/y^ with upregulation of miR-199a-1.		[177]
KI-DNM2R465W (DNM2^RW/+^)	Point mutation A>T in exon 11.		Failed to reproduce the autosomal-dominant form of human CNM.	Neurotransmission is maintained and the mutation shows spatial and temporal muscle involvement as in the similar human mutations.	Homozygous mice show neonatal lethality. The level of central nuclei in muscle fibers is much lower (10%) in homozygous mice than seen in patients (up to 90%).	[178]
miR-133a dKO	Double mutation, missing miR-133a-1 and miR-133a-2.			Upregulation of dynamin 2. Slowly developing CNM.	CNM only in type II fibers of mice, in contrast to the type I fiber predominance in human DNM2 patients.	[179]
DNM2^+/–^	Target exon 8, heterozygous.	DNM2^–/–^ mice are embryonically lethal.	Homozygous mice are embryonically lethal.	DNM2^+/–^ mouse is physiologically and clinically similar to WT mouse. Differences in muscle function were not detectable.		[176]
BIN1^–/–^ DNM2^+/–^	BIN1 KO mouse that is heterozygous for DNM2. Floxed exon 20.	Similar mutations were found in CNM patients.		Similar to WT.		[180]
DNM2^SL/+^	Mouse harboring the S619L DNM2 mutation.	Mimics the S619L missense human mutation.		An early and severe motor defect linked to force reduction and mitochondria structural anomalies	Centralization of nuclei is less prominent in adult mice.	[181]

**Table 6 ijms-21-08935-t006:** Mouse models for malignant hyperthermia.

Model System	Genetic Changes in the Mouse Model/Mutation	Genetic Similarity/Genetic Background		Likeliness of Phenotype and Symptoms		References
		Advantage	Disadvantage	Advantage	Disadvantage	
Y524S RyR1 knock-in mice	Missense mutation in the RyR1 gene.	Exact genetical similarity to a human mutation.	Homozygous mice die at the early stage of intrauterine life (17th day) or soon after birth.	100% similar to the human phenotype.	none	[230,243,244]
R163C RyR1 knock-in mice	Missense mutation in the RyR1 gene.	Exact genetical similarity to a human mutation.	Homozygous mice are not viable at birth.	100% similar to the human phenotype.	none	[230,244]
G2435R RyR1 knock-in mice	Missense mutation in the RyR1 gene.	Exact genetical similarity to a human mutation. Both homozygous and heterozygous mice are viable and fertile.	Some homozygous males died spontaneously.	100% similar to the human phenotype.	none	[245]
T4826I RyR1 knock-in mice	Missense mutation in the RyR1 gene.	Exact genetical similarity to a human mutation. Heterozygous or homozygous mice both survive.	None.	100% similar to the human phenotype.	none	[172]

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
