# Peer review of "From Mice to Humans: An Overview of the Potentials and Limitations of Current Transgenic Mouse Models of Major Muscular Dystrophies and Congenital Myopathies"

_ijms, 2020, doi:10.3390/ijms21238935_

Round 1

Reviewer 1 Report

This review by Sztretye et al. summarizes and provides essential information about murine models for the study of muscular dystrophies and congenital myopathies. The presentation of this review is necessary and opportune.  As far as this reviewer is concerned, the information compiled in this review will help the muscle researcher interested in disease mechanisms and the translational researcher aiming to develop new diagnostic and therapeutic avenues.

The corresponding and senior authors are well-known muscle biomedical scientists and experts on skeletal muscle excitation-contraction coupling and intracellular Ca2+ signaling.  In general, the manuscript is well written and easy to follow. However, several minor issues require attention before a final approval decision can be made.

*General recommendations, typos, and minor issues.

Figure 1.  Change the color of DNM2 labels to a darker yellow.  Increase the size of the nucleus.

  1. Page 2, line 91, delete 'or'.
  2. Page 4, line 152 'has been' replace with 'was'.
  3. Page 4, line 172, define 'WD': i.e., tryptophan and aspartic acid (WD) repeats.
  4. Page 4, line 182, 'has been' replace with 'was' or rephrase this sentence.
  5. Page 5, line 211, briefly define 'T tubules'.
  6. Page 6, line 255, fix quotation marks.
  7. Page 6, line 258, use 'characteristic' instead of 'labeled'.
  8. Page 7, line 304. Delete 'as of today' and replace it with: To the best of our knowledge, at the time of writing this review, there are…
  9. Page 7, line 319, Define 'MH'.
  10. Page 8, line 353, delete 'a lot of' replace with 'multiple'.
  11. Page 8, there is no reference to Table 4 in the text in section 3.2. You may add it at the end of line 356.
  12. Page 8, lines 360-363, revise or split this sentence. It is not clear, consider: The cause of the disease is mutations in the MTM1 gene encoding the universal phosphatase myotubularin, which plays a role in…
  13. Page 9, line 399: 'revised'; do the authors mean 'reverted'?
  14. Page 9, line 417, include a brief description of what tamoxifen is: i.e., Tamoxifen, a selective estrogen receptor modulator used in breast cancer therapy, …
  15. Page 10, line 467. 'BIN1 with DNM2' replace with 'BIN1 and DNM2'.
  16. Page 11, line 470. Typo. 'Hypotised' should read 'hypothesized'.
  17. Page 11, lines 509-510, delete 'in their other paper'.
  18. Page 12, line 561, delete 'On the other hand' replace with: 'Yet, one has to…'

Author Response

We thank the reviewer for the careful reading of the manuscript and the constructive remarks. We have made the changes suggested to improve and clarify the manuscript. Please find below a detailed point-by-point response to all comments (reviewers’ comments in Italics, our replies in black).

Figure 1. Change the color of DNM2 labels to a darker yellow. Increase the size of the nucleus.

The figure has been redrawn taking into account the reviewers’ suggestions.

Page 2, line 91, delete 'or'.

Page 4, line 152 'has been' replace with 'was'.

Page 4, line 172, define 'WD': i.e., tryptophan and aspartic acid (WD) repeats.

Page 4, line 182, 'has been' replace with 'was' or rephrase this sentence.

Page 5, line 211, briefly define 'T tubules'.

Page 6, line 255, fix quotation marks.

Page 6, line 258, use 'characteristic' instead of 'labeled'.

Page 7, line 304. Delete 'as of today' and replace it with: To the best of our knowledge, at the time of writing this review, there are…

Page 7, line 319, Define 'MH'.

Page 8, line 353, delete 'a lot of' replace with 'multiple'.

Page 8, there is no reference to Table 4 in the text in section 3.2. You may add it at the end of line 356.

Page 8, lines 360-363, revise or split this sentence. It is not clear, consider: The cause of the disease is mutations in the MTM1 gene encoding the universal phosphatase myotubularin, which plays a role in…

Page 9, line 399: 'revised'; do the authors mean 'reverted'?

Page 9, line 417, include a brief description of what tamoxifen is: i.e., Tamoxifen, a selective estrogen receptor modulator used in breast cancer therapy, …

Page 10, line 467. 'BIN1 with DNM2' replace with 'BIN1 and DNM2'.

Page 11, line 470. Typo. 'Hypotised' should read 'hypothesized'.

Page 11, lines 509-510, delete 'in their other paper'.

Page 12, line 561, delete 'On the other hand' replace with: 'Yet, one has to…'

Thank you for pointing out these; all of the above were corrected in the revised manuscript thus improving to quality of the writing.

Also, please note that following the suggestion of the other reviewer we have consulted with a neurologist from the University of Debrecen and decided to incorporate a new sub-chapter: 2.3. Facioscapulohumeral dystrophy (FSHD); consequently a new Table summarizing the mouse models used for the study of FSHD was also included.

Reviewer 2 Report

In this paper from Sztretye and coalleagues made an extensive revision of the main muscular dystrophies and congenital myopathies and their animal models is presented. I think the content of this article is excellent, it summarizes in a very easy way the more important mainly mouse models of each one of these diseases and their derived knowledge in mechanisms or assayed therapeutic interventions. The paper is well written, easy to follow and tables are very helpful in illustration the big collection of animal models for each one of these diseases. In my opinion this paper deserves publication but some minor changes are needed.

  1. As the paper only can summarize the “main” muscular dystrophies and congenital myopathies, this should be somehow illustrated in the tittle. Otherwise some readers would expect an exhaustive overview of all muscular dystrophies and would be disappointed.
  2. I am not a clinician and I cannot discern if really all the main muscle dystrophies and congenital myopathies are included in this review, but I would appreciate if someone with this expertise can decide if something is lacking.
  3. Figure 1 looks a little bit old and with a style that it is not fitting with the elegant content of the writing. Some words in yellow are difficult to read, but in general I would try to make a more modern version of it.
  4. In line 399 revised needs to be replaced by reversed.

Author Response

We thank the reviewer for the careful reading of the manuscript and the constructive remarks. We have made the changes suggested to improve and clarify the manuscript. Please find below a detailed point-by-point response to all comments (reviewers’ comments in Italics, our replies in black).

1. As the paper only can summarize the “main” muscular dystrophies and congenital myopathies, this should be somehow illustrated in the tittle. Otherwise some readers would expect an exhaustive overview of all muscular dystrophies and would be disappointed.

We want to thank the reviewer for drawing our attention to this; we have corrected the title of the manuscript to make it clear for the reader what the topic of the review is.

2. I am not a clinician and I cannot discern if really all the main muscle dystrophies and congenital myopathies are included in this review, but I would appreciate if someone with this expertise can decide if something is lacking.

Thank you for pointing this out. We have consulted with a neurologist from the University of Debrecen and decided to incorporate a new sub-chapter about 2.3. Fascioscapulohumeral dystrophy (FSHD) which is the third most common muscular dystrophy; consequently a new Table summarizing the mouse models used for the study of FSHD was also included.

3. Figure 1 looks a little bit old and with a style that it is not fitting with the elegant content of the writing. Some words in yellow are difficult to read, but in general I would try to make a more modern version of it.

The figure has been redrawn taking into account the reviewers’ suggestions.

4. In line 399 revised needs to be replaced by reversed.

Done.